

# Development of a portable laser-flash photolysis Faraday rotation spectrometer for measuring atmospheric total OH reactivity

Bo Fang[1], Nana Wei[1], Weixiong Zhao[1], Nana Yang[1], Hao Zhou[1], Heng Zhang[1,2], Jiarong Li[1], Weijun Zhang[1], Yanyu Lu[3,4], Zhu Zhu[4,5], Yue Liu[4,5]

[1]Laboratory of Atmospheric Physico-Chemistry, Anhui Institute of Optics and Fine Mechanics, HFIPS, Chinese Academy of Sciences, Hefei 230031, Anhui, China
[2]Institutes of Physical Science and Information Technology, Anhui University, Hefei 230039, Anhui, China
[3]Anhui Institute of Meteorological Sciences, Anhui Province Key Laboratory of Atmospheric Science and Satellite Remote Sensing, Hefei 230031, Anhui, China
[4]Shouxian National Climatology Observatory, Huaihe River Basin Typical Farm Eco-meteorological Experiment Field of CMA, Shouxian 232200, Anhui, China
[5]Anhui Shouxian Meteorological Bureau, Shouxian 232200, Anhui, China

*Correspondence to*: Weixiong Zhao (wxzhao@aiofm.ac.cn), Weijun Zhang (wjzhang@aiofm.ac.cn)

**Abstract.** Quantitative measurements of atmospheric total OH reactivity ($k_{OH}'$) provide crucial insights into atmospheric photochemistry. However, widespread application of total OH reactivity measurements is challenging due to insufficient equipment and the complexity of existing instrumentation. In this work, we report the development of a portable laser-flash photolysis Faraday rotation spectroscopy (LP-FRS) instrument for real-time and in-situ measurement of $k_{OH}'$. To achieve efficient overlapping between the pump and probe laser and realize a long effective absorption path length, thus enabling high sensitivity measurement, a specific Herriott-type pump-probe optical multi-pass cell was designed with an overlapping factor of up to 75.4%. The instrument's optical box dimensions were 130 cm $\times$ 40 cm $\times$ 35 cm. The obtained efficient absorption path was ~ 28.5 m in a base length of 77.2 cm. The $k_{OH}'$ detection precisions of the LP-FRS instrument were 2.3 s$^{-1}$ and 1.0 s$^{-1}$ with averaging times of 60 s and 300 s, respectively. The $k_{OH}'$ measurement uncertainty was evaluated to be within 2 s$^{-1}$. Field measurement was performed, and the difference between the measured $k_{OH}'$ and the model simulated from the measured reactive species was analysed. The developed portable LP-FRS instrument extends the measurement methods of atmospheric total OH reactivity, and has certain advantages in cost, operation, and transportation, which will play an increasingly important role in future atmospheric chemistry research.

## 1 Introduction

The hydroxyl (OH) radical is the most important oxidant in the atmosphere during daytime. It initiates the oxidation of most natural and anthropogenic trace gaseous species, thereby dominates their atmospheric lifetime. Knowledge of tropospheric OH chemistry contributes to our understanding of air pollution and climate change (Lu et al., 2018; Nicely et al., 2018). However, due to the large number ($10^4$ - $10^5$) of volatile organic compounds (VOCs) (Goldstein and Galbally, 2007), a comprehensive interpretation of the sink mechanisms of OH is extremely challenging.



Total OH reactivity ($k_{OH}'$), the inverse of the OH chemical lifetime ($\tau_{OH}$), serves as an crucial parameter for estimating the total loss rate of OH due to all atmospheric OH reactants (Yang et al., 2016). It is defined as the sum of OH reactant

concentrations ([X]) weighted by their reaction rate coefficient with OH ($k_{OH+X}$), which can be expressed as

$$k_{OH}^{'} = \tau_{OH}^{-1} = \sum k_{OH+X_i}[X_i] \tag{1}$$

where $X_i$ is the $i$-th reactant. Measurement of $k_{OH}'$ provides a powerful tool for both field campaigns and laboratory studies in atmospheric photochemistry (Stone et al., 2012; Fuchs et al., 2013). The balance between OH production and loss rate provides additional information on the OH sources (Martinez, 2003; Hens et al., 2014). The difference between measured

and calculated $k_{OH}'$ can be used to estimate the contribution from individually measured species, the total missing reactivity, and the role of unknown VOCs (Di Carlo et al., 2004; Mao et al., 2010; Sinha et al., 2012). The measured total OH reactivity can also be used as the chemical closure of the reactive carbon budget (Hunter et al., 2017; Safieddine et al., 2017; Heald et al., 2020). In addition, $k_{OH}'$ measurements help to estimate instantaneous production potential and production regime of ozone (Sinha et al., 2012; Li et al., 2021; Kohno et al., 2022; Li et al., 2022). Recently, the long-term trend of $k_{OH}'$ has been

proven to be a key atmospheric oxidation capacity parameter for the formulation of ozone ($O_3$) pollution mitigation strategy (Wang et al., 2023).

Several methods for OH reactivity measurement have been developed (Yang et al., 2016; Fuchs et al., 2017), which can be divided into three categories: the indirect method known as the comparative reactivity method (CRM) (Sinha et al., 2008); the semi-direct technique involving flow tube - chemical ionization mass spectrometry (FT-CIMS) (Muller et al., 2018); and

the direct method, including flow tube - laser induced fluorescence (FT-LIF) (Kovacs and Brune, 2001; Mao et al., 2009; Ingham et.al., 2009; Hansen et al., 2014) and laser-flash photolysis - laser induced fluorescence (LP-LIF) (Sadanaga et al., 2004; Lou et al., 2010; Parker et al., 2011; Stone et al., 2016).

The CRM indirectly determines $k_{OH}'$ by the competitive kinetics for OH between a reference molecule not present in normal atmospheric condition (e.g., pyrrole) and all reactive atmospheric species in ambient air. Commercial proton-

transfer-reaction mass-spectrometry (PTR-MS) or gas chromatography (GC) is employed to detect the concentration change of pyrrole (Nölscher et al., 2012). The direct method determined $k_{OH}'$ from the measured time-dependent OH decay. The FT-LIF directly measures OH decay by controlling the reaction time through the movement of an OH injector along a flow tube. A LIF instrument is positioned downstream of the flow tube to monitor the OH concentration signal intensity. In LIF instrument, the sample is drawn into a low-pressure (~ 1.5 Torr) cell via gas expansion. A 308 nm dye laser is used to excite

OH, and the resulting 308 nm fluorescence emitted by OH is collected for concentration evaluation. The LP-LIF is a pump-probe technique where OH decay can be observed with high time resolution after each flash without needing to determine the reaction time. In this technique, OH is produced by laser-flash photolysis of $O_3$ at 266 nm in the presence of water vapour, making it less susceptible to the recycling process caused by nitric oxide (NO) compared to the above instruments using water vapour photolysis (Sadanaga et al., 2004; Lou et al., 2010). In the semi-direct technique of FT-CIMS, sulphuric





acid ($H_2SO_4$) instead of OH is measured by a CIMS instrument to record the data point of OH decay at each reaction time. The reaction time can be varied by converting OH to $H_2SO_4$ at different fixed positions within the flow tube.

The instrument performances of the above techniques have been intercompared and validated in the simulation chamber SAPHIR, demonstrating that the direct methods offer advantages in detection precision and accuracy (Fuchs et al., 2017). However, the high cost of development and operation, limited instruments, complex operation and calibration procedures,

and relatively large size of these instruments hinder the widespread application of measuring OH reactivity.

In this work, we report the development of a portable LP-FRS instrument for total OH reactivity measurement. The time-resolved laser-flash photolysis Faraday rotation spectroscopy (LP-FRS) is a novel technique that employs mid-infrared semiconductor diode laser as the probe laser for $k_{OH}'$ measurement (Wei et al., 2020). Since FRS relies on the detection of the probe light polarization state rotation induced by paramagnetic molecules in a longitudinal magnetic field, the laser noise

and molecule interferences are significantly reduced, which enables the FRS system to directly, highly sensitive, and absolutely monitor the concentration of OH without any chemical interferences (Litfin et al., 1980; Zhao et al., 2018). The dimensions of the developed instrument were 130 cm $\times$ 40 cm $\times$ 35 cm. The achievable detection precision of $k_{OH}'$ was 1.0 s[-1] with 300 s averaging time. Field test in a suburban area was performed to demonstrate the capability of the LPF-FRS instrument.

**2 Experimental setup**

A schematic diagram of the developed LP-FRS instrument is given in Fig.1(a). The instrument comprises a mid-infrared FRS system for direct measurement of OH and an ultraviolet (UV) laser-flash photolysis system for generating OH. The probe light and the UV beam have an overlapping in an Herriott-type optical multi-pass cell (MPC), enabling simultaneous monitoring of OH by the FRS system during the generation and reaction with reactants. Optical components from both

systems are integrated into a single unitary box, with all communications and gas tubes connected to designated interfaces. The dimensions of the box are 130 cm $\times$ 40 cm $\times$ 35 cm, making the LP-FRS instrument portable for field applications.





**Figure 1:** (a) Schematic diagram of the developed laser-flash photolysis Faraday rotation spectrometer (LP-FRS) which
consists of a mid-infrared Faraday rotation spectroscopy system and a laser photolysis system. The OH radicals are
generated by laser-flash photolysis at 266 nm in a Herriott cell wound with copper wires, and detected simultaneously by
Faraday rotation spectroscopy via the overlapping mid-infrared optical paths. P, polarizer; FM, foldable mirror; SM, silver
mirror; L, lens; PD, photodetector; YM, Nd: YAG mirror; BE, beam expander; DAQ, data acquisition card. (b) Assembly
diagram of the Herriott-type pump-probe MPC. The coil is wound around the body of the MPC. A water-cooling interlayer
is designed for temperature control of the solenoid coil and the cell.



## 2.1 Mid-infrared Faraday rotation spectrometer

A mid-infrared continuous wave distributed feedback laser (cw-DFB laser, Nanoplus GmbH) emitting at 2.8 μm is used as the probe laser. The current and temperature of the laser chip are controlled by a laser controller (LDC501, Stanford

Research Systems). By changing the injection current from 90 to 130 mA at the operating temperature of 33 °C, the wavelength of the DFB laser can be tuned from 3568.939 to 3568.362 cm$^{-1}$. The current tuning coefficient is about 0.0024 cm$^{-1}$/mA near the target Q(1.5e) line of the $^2\Pi_{3/2}$ state of OH at 3568.523 cm$^{-1}$. The selected line has the strongest line strength of $S = 9.032 \times 10^{-20}$ cm$^{-1}$/(molecule cm$^{-2}$) at 296 K (Gordon, et al., 2022) with the largest effective $g_J$ value of 0.936 in the infrared region, which make it preferable for the FRS detection (Zhao et al., 2011; Zhao et al., 2012). The collimated

beam output from the laser head passes through a Rochon prism (Foctek Photonics), with an extinction ratio of $\xi < 5 \times 10^{-6}$, to establish a linearly polarized state, and then incident into a Herriott-type pump-probe MPC. A He-Ne laser serves as an indicator. The beam waists of both lasers are aligned at the centre of the MPC to minimize beam divergence after each reflection (Pilgrim et al., 1997). A flipper optical mount facilitates switching between the two lasers. A second Rochon prism is placed at the output path to analyse the polarization state. The exited beam from the MPC is focused on a

thermoelectrically cooled mercury cadmium telluride (MCT) photodetector (PVI-4TE-3.4, VIGO System). To effectively modulate the magnetic circular birefringence in a static magnetic field (Zhao et al., 2018; Fang et al., 2020; Wei et al., 2020), a 33 kHz sinusoidal wave from a lock-in amplifier (SR830, Stanford Research) is added to the laser injection current. The detector signal is processed by the lock-in amplifier to demodulated the second harmonic (2$f$) of the FRS signal. As the laser current is fixed at the absorption peak of the OH radical, time-resolved Faraday rotation spectrum that directly reflects the

concentration variation of OH radical can be measured (Wei et al., 2020; Cheng et al., 2023).

The pump-probe MPC, as shown in Fig.1(b), consists of a cylindrical stainless steel tube with an inner diameter of 5 cm, a total length of approximately 89 cm, and a sample volume of 1.5 L. At both ends, a pair of 6.8 cm diameter calcium fluoride (CaF$_2$) windows are used for sealing and light transmission. Two gold-coated concave spherical mirrors, each 5 cm in diameter, are spaced 77.2 cm apart within the cell. Each mirror features a 5 mm diameter hole for probe light incidence

and exit, and a central 32 mm diameter hole (i.e., the maximum diameter permissible for the UV beam passage) for the expanded photolysis beam. Mirror tilt and spacing are adjustable via three screws distributed circularly on the mirror mount. At each end of the MPC, there are eight circular quartz observation windows near the mirrors to facilitate multi-pass light adjustments. 25 reflection spots arrange on the mirror surfaces in a circular pattern with 2 cm radius. The total path length of the MPC is 37.8 m.

A solenoid coil, wrapped with 1 mm diameter red-copper enamelled wires, is wound around the stainless steel tube and operates in DC mode to offer a static magnetic field for FRS. The length and the outer diameter of the coil are 59 cm and 10 cm, respectively. The magnetic intensity tuning coefficient at the centre of the coil is 73 G/A. A water-cooling interlayer is designed for temperature control of both the solenoid coil and the cell.



## 2.2 Ultraviolet laser-flash photolysis system

A flashlamp pumped Nd: YAG laser (Big Sky Laser Ultra 100, Quantel) is employed as the photolysis laser. The laser wavelength is frequency doubled to generate the fourth harmonic radiation at 266 nm with pulse energy of 25 mJ, energy stability of ~ 2%, pulse length of ~ 6 ns, and beam diameter of ~ 4 mm. The water cooled laser head has a size of ~ 30.6 cm × 7.6 cm × 5.6 cm, and is controlled by an integrated cooling and electronics unit. The 266 nm pulse emitted from the laser head is directed coaxially into the MPC using two dielectrically coated 1-inch diameter mirrors. Prior to entering the cell, the

diameter of the 266 nm beam is expanded to 32 mm using a beam expander consists of a quartz concave lens with a focal length of 12.5 mm and a quartz convex lens with a focal length of 100 mm.

A digital delay generator (DG645, Stanford Research Systems) was used to control the time sequence of laser-flash photolysis to record the time-resolved OH decay curve. The pump and Q-switch of the Nd: YAG laser are synchronized with two 4 Hz TTL (transistor-transistor logic) pulses delayed by 30 ms relative to the data acquisition to achieve the baseline of

the OH decay curve. The rising edges of the pulses are used for triggering. The spectrum is sampled with 1000 data points, each separated by a time interval of 0.2 ms.

## 3 Instrument performance

### 3.1 Characterization of the pump-probe MPC

The MPC in the instrument determines the effective absorption path length of the FRS system for OH measurement (Wei et

al., 2020; Yan et al., 2020). As shown in Fig.2(a), the mid-infrared light undergoes multiple passes between two mirrors within the MPC is used for OH detection; the expanded 266-nm UV pulse is employed for producing OH. The overlapping factor ($\eta = l/d$) can be defined as the ratio of the overlapping length ($l$) to the base length ($d$) of the MPC. For developing portable instruments, increasing the overlapping factor is crucial to achieve long effective path length while reducing the MPC base length. The Herriott-MPC in the developed LP-FRS instrument is specially designed with a small multi-pass light

distribution circle radius at the centre to achieve a high overlapping factor. The radius of the multi-pass light distribution circle at the centre of the Herriott-MPC can be calculated with (Trutna and Byer, 1980):

$$r_c = r \left( \frac{g_1 + 2g_1 g_2 + g_2}{4 g_2} \right)^{1/2} \tag{2}$$

where g ($g = g_1 = g_2 = \cos\theta = 1 - d/R$) is the parameter that describes the optical resonance stability of optical cavity or MPC, $\theta$ is half of the angel between two adjacent reflection light points on mirror surface, $d$ is the base length, $R_1 = R_2 = R$ is

the curvature radii of the mirrors, $r$ is the radius of the spot distribution circle.

Fig.2(b) illustrates the variations in $r_c$, g and $\eta$ as functions of $\theta$. When $\theta$ value is below than 79.2 °, the $r_c$ exceeds the 16 mm radius of the photolysis beam, resulting in no overlapping path. As the $\theta$ approaches -1, the value of $r_c$ decreases and the





$\eta$ value increases. Considering the difficulties in processing and testing mirror curvature, $\theta$ was set to 158.4 °, yielding an $r_c$ value of 3.7. The achieved $\eta$ was up to ~ 75.4%, corresponding to an overlapping path length of ~ 28.5 m.


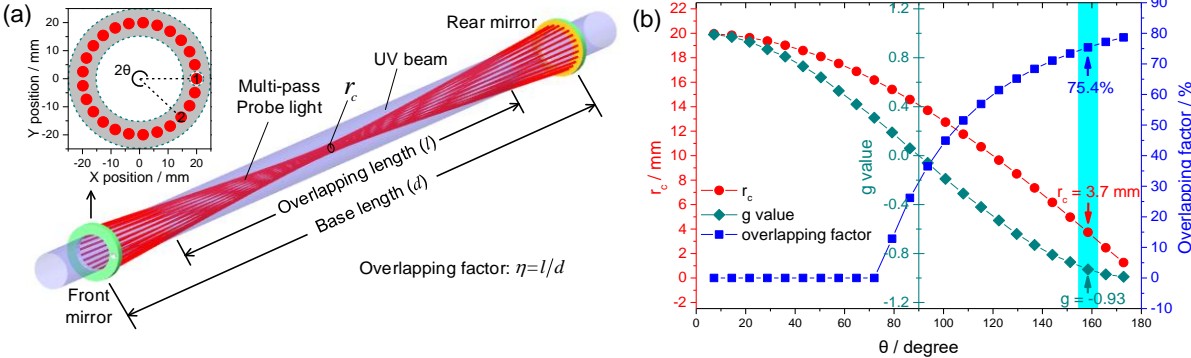

**Figure 2:** (a) Schematic of the Herriott multi-pass cell and the beam pattern on mirror surface for pump-probe; (b) multi-pass light waist ($r_c$), g value and overlapping factor ($\eta$) as functions of $\theta$. $\theta$ is half of the angel between two adjacent reflection spots. The $\theta$ of 158.4 °is selected, achieving an $r_c$ = 3.7 mm and a $\eta$ of 75.4%.


**Table 1.** Performance comparison of MPCs used for laser photolysis in pump-probe techniques.

| MPC type | Base length (cm) | Effective overlapping path length (m) | Overlapping factor | References |
|---|---|---|---|---|
| Multi-pass arrangement | 150 | 4.4 | 35.9% | Lewis et al., 2018 |
| | 150 | 7.0 | 41.9% | Lewis et al., 2018 |
| Herriott cell | 90 | 12 | 33.3% | Pilgrim et al., 1997 |
| | 125 | 45 | 64.0% | Pilgrim et al., 1997 |
| | 100 | 3.5 | 18.4% | Qian et al., 2000 |
| | 73 | 13 | 80.2% | Luo et al., 2019 |
| | 65.5 | 24.5 | 58.6% | Luo and Horng, 2020 |
| | 65.5 | 20.9 | 50% | Luo et al., 2020 |
| | 122 | 25 | 41.7% | Wei et al., 2020; Cheng et al., 2023 |
| | 91.3 | 6.3 | 32.9% | Yan et al., 2020; Teng et al., 2021 |
| | **77.2** | **28.5** | **75.4%** | **This work** |



A comparison of effective overlapping path lengths and overlapping factors with literature reported MPCs used in pump-probe techniques is shown in Table 1. The effective overlapping path length of these MPCs ranged from sever meters to tens

of meters with base lengths around 100 cm (Pilgrim et al., 1997; Luo et al., 2019; Qian et al., 2000; Lewis et al., 2018; Luo and Horng, 2020; Luo, 2020; Wei et al., 2020; Yan et al., 2020; Teng et al., 2021; Cheng et al., 2023). The overlapping factor of our MPC was comparable to that developed by Lou et al., 2019, for a similar base length, while our effective overlapping path length was twice as long.

### 3.2 Optimization of the FRS system

For weak absorption, the FRS signal ($S_{FRS}$) and total noise ($N_{tot}$) of the system can be expressed as functions of the analyser offset angle ($\phi$) from the crossed polarization of the light (Zhao et al., 2011; Wei et al., 2020):

$$S_{FRS}(\nu) = \gamma N S L P_0 \sin(2\phi)\chi(\nu) \tag{3}$$

$$N_{tot}(\phi) = \sqrt{N_0^2 + \left(N_1\sqrt{\left(\sin^2(\phi) + \xi\right)}\right)^2 + \left(N_2\left(\sin^2(\phi) + \xi\right)\right)^2} \tag{4}$$

where $\nu$ is the laser frequency, $\gamma$ is the instrumentation factor, $N$ is the OH concentration, $S$ is the absorption line strength

of OH, $\chi$ is Faraday rotation lineshape (Westberg and Axner, 2014). $N_0$, $N_1\sqrt{\left(\sin^2(\phi)+\xi\right)}$ and $N_2\left(\sin^2(\phi)+\xi\right)$ are the detector noise, shot noise and laser noise of the system, respectively. It is noted that FRS signal reaches its maximum value when $\phi = \pm45$°, while the total noise is more sensitive to $\sin^2(\phi)$. The maximum signal-to-noise ratio (SNR) usually occurs at a small offset angle, which depends on system noise (Lewicki et al., 2009).

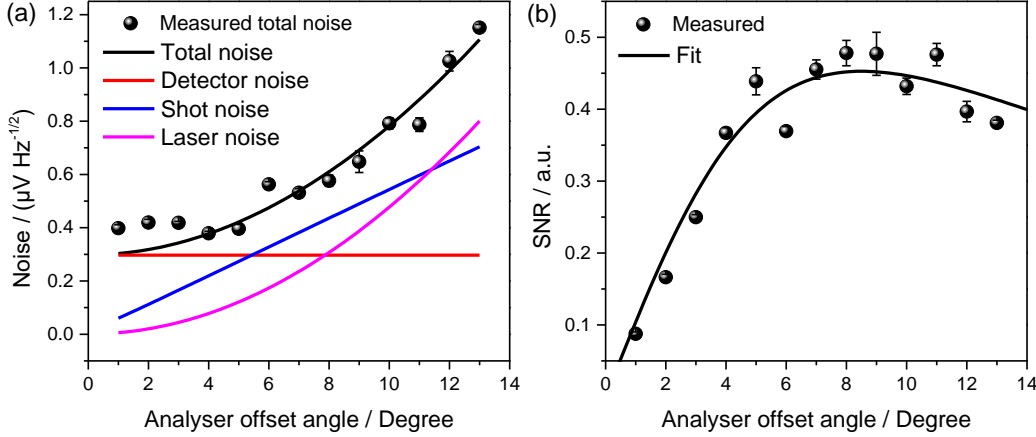

**Figure 3:** (a) Nosie and (b) signal-to-noise ratio analysis of the FRS system. The optimum offset angle of analyser is 8 °.


The total noises at various offset angles and the detector noise were measured with the lock-in amplifier to determine the noise sources and the optimum angle ($\phi_{opt}$) for maximizing the SNR of the FRS system (Zhao et al., 2011). Fig.3(a) shows a



fit analysis of the data using Eq.(4). For offset angles below 5.5 °, detector noise is the main noise of the system. The measured detector noise was 0.30 μV Hz$^{-1/2}$ which closely agreed with the manufacturer's specified value of 290 nV Hz$^{-1/2}$. Laser noise dominates and rapidly increased beyond an offset angle greater of 11.5 °. Since FRS signal is proportional to $\sin(2\phi)$, the relative SNR for a given absorption at a fixed laser frequency can be evaluated from $SNR \propto \sin(2\phi)/N_{tot}$. As shown in Fig.3(b), the SNR of our system peaks at the $\phi_{opt}$ of ~ 8 °. At this angle, the laser noise was suppressed to 0.31 μV

Hz$^{-1/2}$, equivalent to the detector noise level. The total system noise was 0.61 μV Hz$^{-1/2}$ which was 1.4 times higher than the measured shot noise of 0.44 μV Hz$^{-1/2}$.

In this work, the magnetic circular birefringence of OH was effectively modulated with wavelength modulation in a static magnetic field generated by the DC coil. The modulation amplitude and magnetic field strength are critical parameters that affecting the intensity of the demodulated FRS signal (Zhao et al., 2018; Fang et al., 2020). The theoretical optimum

modulation amplitude is 2.2 times of the HWHM (half width at half-maximum) of the absorption lineshape (Schilt et al., 2003). The optimum magnetic field strength ($B_{opt}$) is the value that can make the Zeeman splitting comparable with the HWHM (Brecha et al., 1997). A direct and effective approach for determine the two optimum parameters is recording the signal intensity values under series amplitudes of the sinusoidal wave output from the lock-in amplifier and different coil currents. As shown in Fig.4, the FRS signal intensity value is calculated from the difference before and after laser-flash. The

maximum intensity occurred at an amplitude of 360 mV and a coil current of 4.2 A, which corresponding to a wavelength modulation amplitude of ~ 0.048 cm$^{-1}$ (~ 2.45 times of the calculated OH absorption linewith of ~ 0.020 cm$^{-1}$ in air) and a $B_{opt}$ of 307 Gauss, respectively.

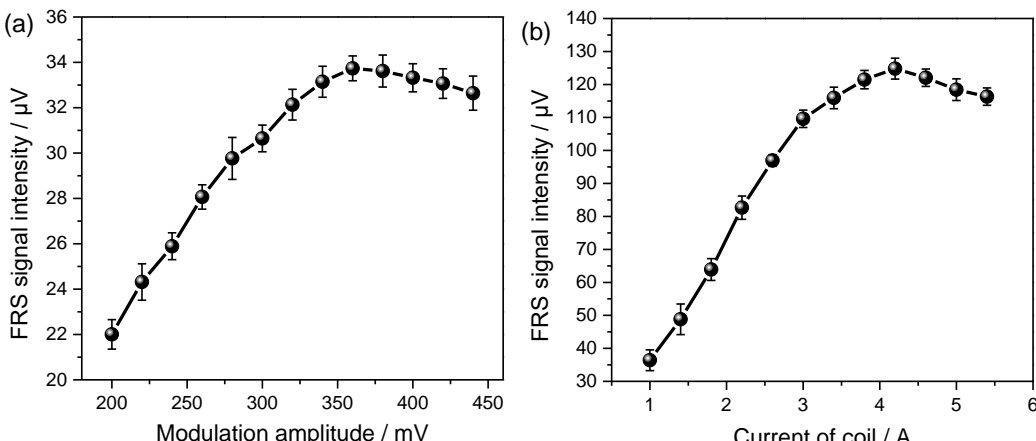

**Figure 4:** FRS signal intensity as functions of (a) modulation amplitude and (b) current of coil. The optimum modulation amplitude and magnetic field strength are 360 mV$_{rms}$ and 307 Gauss, respectively.




### 3.3 OH concentration in the cell

To produce sufficient OH in the MPC, a small flow rate (~ 0.25 L/min) of zero air which passed through a UV lamp and a
bubbling bottle to generate $O_3$ and water vapour, is added to the main sampling flow (~ 6.0 L/min), resulting in a total flow
rate of 6.25 L/min. The operation pressure in the MPC is set to 200 mbar, and maintained by a butterfly valve (DN40, VAT).
The flow velocity in the cell is 27 cm/s, corresponding to a Reynolds number of ~ 170, which met the laminar flow condition.
OH is produced by the 266 nm laser photolysis of $O_3$, then followed by reaction of $O(^1D)$ with water vapour (Sadanaga et al.,
2004):

$$O_3 + h\nu(\lambda = 266nm) \rightarrow O_2 + O(^1D) \tag{R1}$$

$$O(^1D) + H_2O \rightarrow 2OH \tag{R2}$$

The concentrations of $O_3$ and the water vapour directly influence the OH producing. A dewpoint sensor was used to
measure the water vapour mixing ratio of the total flow. Based on Magnus-Tetens formula (Lawrence, 2005), the absolute
water vapour mass concentration can be evaluated from $\rho(H_2O)=e/(R_w \cdot T)$, where $R_w$ = 461.52 J/(kg K), $T$ is the sample
temperature, $e = C \times 10^{(A \cdot T_d/(B+T_d))} \cdot (P/P_0)$ is the actual water vapour pressure at current pressure of $P$, $A$ = 17.625, $B$ =
243.04 °C, $C$ = 610.94 Pa, $P_0$ is the standard atmospheric pressure, $T_d$ is the measured dewpoint temperature. The calculated
water vapour volume concentration was ~ 0.13% when using zero air for system test, while it increased to ~ 1.5% when
measuring real atmosphere. The $O_3$ concentration is estimated by measuring the energy of the photolysis laser pulse with the
UV lamp on and off. Only about 0.3% of the pulse energy was absorbed by $O_3$. Based on the Beer-Lambert law and the $O_3$
absorption cross section of $\sigma = 9.65 \times 10^{-18}$ cm$^2$ at 266 nm (Sadanaga et al., 2004), the $O_3$ concentration is determined to be ~
800 ppbv. The recommended quantum yield of the $O(^1D)$ produced by laser flash photolysis is ~ 0.9 (Atkinson et al., 2004),
resulting in ~ $2.2 \times 10^{11}$ molecule/cm$^3$ of $O(^1D)$. The number density of OH produced by the $O(^1D)$ can be estimated by
(Wei et al., 2020):

$$[OH] = \frac{2k_6[O(^1D)][H_2O]}{(k_5[M]+k_6[H_2O])[O(^1D)]} \cong \frac{2k_6}{k_5}\chi_{H_2O} \tag{5}$$

where [OH], [O($^1$D)], [$H_2O$] and [M] represent the number densities of the corresponding molecule. M is the "bath" gas
during the chemical reaction of OH formation. $\chi_{H_2O}$ is the volume concentration of water vapour. $k_5$ and $k_6$ are 2.9 $\times 10^{-11}$
cm$^3$ molecule$^{-1}$ s$^{-1}$ and 2.2 $\times 10^{-10}$ cm$^3$ molecule$^{-1}$ s$^{-1}$ at 298 K, respectively. The concentration of OH produced in the cell
was ~ $4.3 \times 10^9$ molecule/cm$^3$ during system test and was ~ $5 \times 10^{10}$ molecule/cm$^3$ during field application due to different
$\chi_{H_2O}$ in the sample.





## 3.4 Kinetics test

The performance of the LP-FRS instrument in measuring different reaction rates were verified with three well-known reactions (Wei et al., 2020; Yan et al., 2020). Reactants from the cylinders (i.e., $CH_4$ (99.999%, Nanjingteqi), CO (2.01%, Nanjingteqi) and NO (100 ppmv, Linde)) were added to the main flow at different flow rates. When the concentrations of reactants are much higher than that of OH, the OH decay rate ($k_{decay}$) follows pseudo-first order kinetics and can be determined by fitting the measured decay spectra to the flowing exponential equation:

$$y = a + b\exp(-k_{decay}t) \tag{6}$$

where $y$ is the FRS signal intensity at the reaction time $t$. $a$ and $b$ are parameters representing the background signal intensity and the initial OH concentration, respectively. Since the fitted values for $y$ and $k_{decay}$ do not depend on the selected time period of the decay curve (Stone et al., 2016), the fit is started at the 180th data point rather than the peak to avoid any fluctuations affecting the fitting result. Fig.5 shows two typical decays with loss rates of $k_{decay}$ = 8.4 $s^{-1}$ and $k_{decay}$ = 50.2 $s^{-1}$ which are given with 60 s averaging time during the measurements. The recorded time-resolved decay spectra clearly depict the entire event including the baseline, the instant generation of OH by laser photolysis, and the decay process.

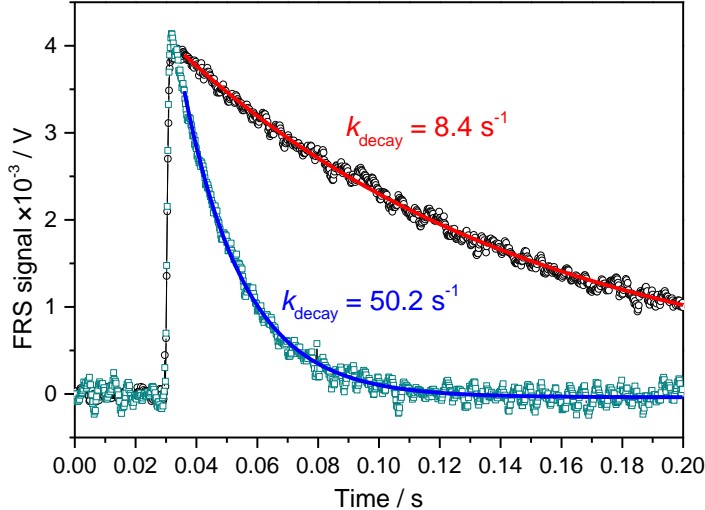

**Figure 5:** Two typical OH decay spectra with different loss rates. Time zero is defined as the moment when the data acquisition trigger occurs.

The OH decay rates in the reactions with three different species were measured. As shown in Fig.6, the reaction rate constants for OH + $CH_4$, OH + CO, and OH + NO at 298 K, obtained from the slope of linear fittings between the decay rates and the concentrations, were found to be $k_{OH+CH4}$ = 6.4 $\times 10^{-15}$ $cm^3$ molecule$^{-1}$ $s^{-1}$, $k_{OH+CO}$ = 1.6 $\times 10^{-13}$ $cm^3$ molecule$^{-1}$ $s^{-1}$, and $k_{OH+NO}$ = 3.0 $\times 10^{-12}$ $cm^3$ molecule$^{-1}$ $s^{-1}$. The slope errors of the fittings were less than 0.1. The measured reaction rate





constants agreement with the IUPAC (International Union of Pure and Applied Chemistry) recommend values of 6.4 (-1.1, +1.3) $\times 10^{-15}$ cm$^3$ molecule$^{-1}$ s$^{-1}$, 1.6 (-0.2, +0.2) $\times 10^{-13}$ cm$^3$ molecule$^{-1}$ s$^{-1}$, and 3.0 (-0.6, +0.8) $\times 10^{-12}$ cm$^3$ molecule$^{-1}$ s$^{-1}$, respectively (Atkinson et al., 2004; Atkinson et al., 2006).


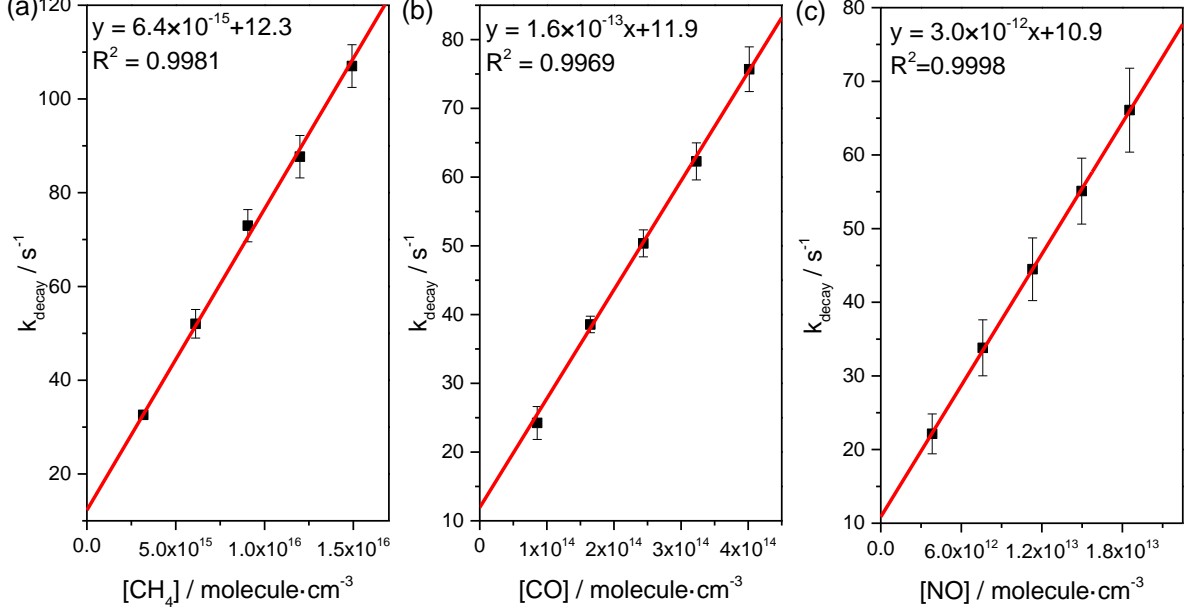

**Figure 6:** Plots of the measured pseudo-first-order rate coefficients vs (a) CH$_4$ concentrations, (b) CO concentrations and (c) NO at 298 K. The measured reaction rate constants are $k_{OH+CH4}$ = 6.4($\pm$0.1) $\times 10^{-15}$ cm$^3$ s$^{-1}$, $k_{OH+CO}$ = 1.6($\pm$0.1) $\times 10^{-13}$ cm$^3$ s$^{-1}$, and $k_{OH+NO}$ = 3.0($\pm$0.1) $\times 10^{-12}$ cm$^3$ s$^{-1}$, which respectively agree with the recommend values.


### 3.5 Precision and uncertainty of $k_{OH}{'}$ measurement

When the LP-FRS instrument is used for measuring atmospheric $k_{OH}{'}$, the fitted $k_{decay}$ value requires corrections for dilution and instrument zero ($k_{zero}$). Incorporating these corrections, the atmospheric $k_{OH}{'}$ is expressed as

$$k_{OH}^{'} = f \times (k_{decay} - k_{zero}) \tag{7}$$

where $f$, given by $f = f_{pressure} \times f_{flow}$, is the total dilution factor which arises from the low operating pressure of 200 mbar and the additional small flow of humidified air containing a constant mixing ratio of O$_3$. $f_{pressure}$ represents the pressure dilution factor given by the ratio of the ambient pressure to the operation pressure, equal to ~ 5. $f_{flow}$ is the flow dilution factor calculated as the ratio of the total flow rate of 6.25 L/min to the sample flow rate of 6.0 L/min, equal to 1.04. Therefore, the total correction factor ($f$) is 5.2.





The instrument zero is critical for calculating $k_{OH}'$ from the observed OH decay rate and is usually assumed constant over a certain observation period. To evaluate instrument zero, OH decays rates were measured in zero air produced by a portable zero gas generator, yielding the $k_{zero}$ of 5.2 s⁻¹. Several factors affect the instrument zero, including the self-reaction of OH, reaction of OH with O₃, OH diffusion, and reaction of OH with residual reactive species in zero air. The self-reaction rate constant of OH is $1.48 \times 10^{-12}$ cm³ molecule⁻¹ s⁻¹ at 298 K (Atkinson et al., 2004), contributing to ~ 1.5% of the instrument

zero at current OH concentration. The reaction rate constant of OH with O₃ is $7.3 \times 10^{-14}$ cm³ molecule⁻¹ s⁻¹ at 298 K (Atkinson et al., 2004), leading to an OH loss rate of 0.3 s⁻¹, which accounts for ~ 6% of the instrument zero. The OH loss rate due to diffusion ($k_{dif}$) under laminar flow condition can be calculated from (Ivanov et al., 2007; Liu et al., 2009):

$$k_{dif} = K_{dif} \times \frac{D_{OH}}{r_{tube}^2} \qquad (8)$$

where $K_{dif} = 3.66$ (for a cylinder cell) is the dimensionless geometric parameter, $r_{tube}$ is the radius of the cell, $D_{OH} = 1.3$ cm²

s⁻¹ is OH diffusion coefficient at 200 mbar and 298 K in air. The calculated OH diffusion loss rate was 0.8 s⁻¹, giving a contribution of ~ 15% of the instrument zero.

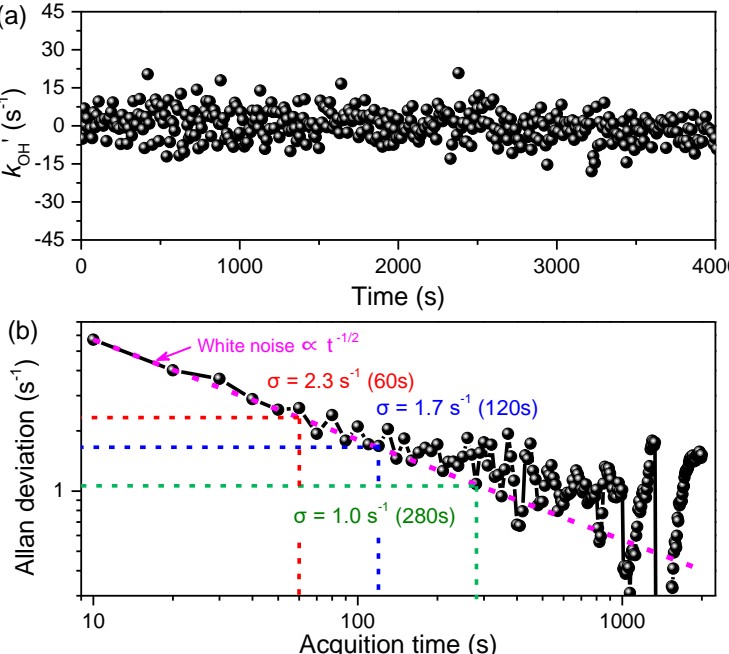

**Figure 7:** (a) Time series of zero air with 8 s time resolution and (b) the Allan deviation analysis of the time series. The measurement precision of $k_{OH}'$ can be improved to 2.3 s⁻¹ and 1.0 s⁻¹ with averaging times of 60 s and 300 s, respectively.





To assess the measurement precision of the LP-FRS instrument, an Allan deviation analysis was conducted on time series of zero air measurements. As shown in Fig.7, the averaging time of $k_{OH}'$ data point was 10 s with total a measurement time of about 4000 s. The measurement precisions of $k_{OH}'$ were 2.3 s⁻¹ and 1.7 s⁻¹ with the acquisition time of 60 s and 120 s, respectively. When the averaging time increased to 300 s, the atmospheric $k_{OH}'$ measurement precision could further improved to 1.0 s⁻¹.

A comprehensive approach to evaluate the instrument's total uncertainty is measuring zero air. Fig.8 illustrates the $k_{OH}'$ values obtained from zero air on different days. Each value was obtained from a one-hour continuous measurements. The time intervals for these zero value measurements are longer than two weeks. The error bars of the data points agreed well with the $k_{OH}'$ measurement precision obtained from Allan deviation analysis. No significant drift was found during measurements, with a mean value close to zero at 0.2 s⁻¹. The $k_{OH}'$ measurement uncertainty of the developed LP-FRS instrument, which can be determined from the deviations of these measurements, was within 2 s⁻¹. The uncertainty of the LP-FRS instrument arises from two main sources: the dilution factor and the instrument zero. The uncertainties associated with the MFCs and the pressure controller used were ~ 1% and < 1%, respectively, resulting in a total uncertainty of the dilution correction factor of less than 2%. The uncertainty of the instrument zero primarily originates from various influencing factors, such as changes in $O_3$ concentrations and residual reactants in zero air.

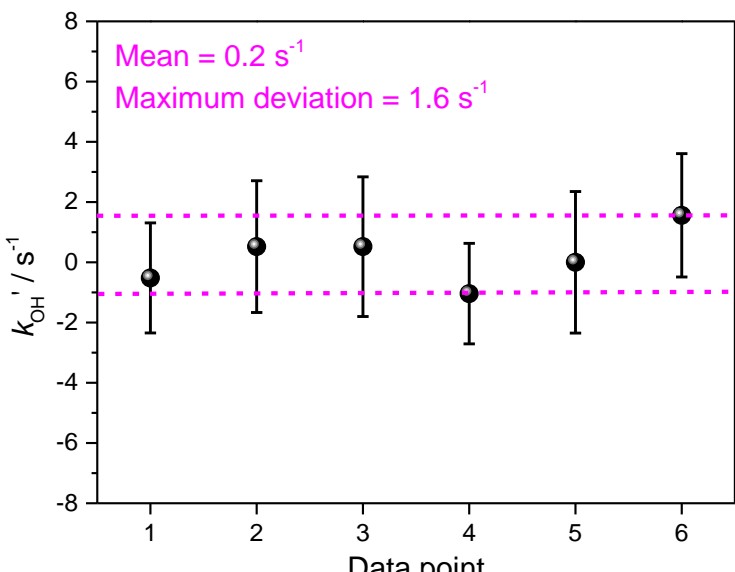

**Figure 8:** Measured $k_{OH}'$ values of zero air on different days. The total instrument uncertainty, determined from the measurements deviations, is within 2 s⁻¹.





## 4 Field performance

The capability of the developed LP-FRS instrument was demonstrated through field measurements of atmospheric total OH reactivity. Measurements were conducted at the park of the Hefei Institutes of Physical Science, Chinese Academy of Sciences (31°91′ N, 117°16′ E) (Khayyam et al., 2024). This observation site is situated on a peninsula in a suburban area, surrounded by water on three sides. There were also an optical and several mechanical processing factories located on the peninsula. The observation period spanned from May 1st to May 4th, 2023.

The LP-FRS instrument was housed in a container with the sampling port positioned ~ 1.2 m above the container's roof. In addition to measuring OH reactivity, concentrations of 115 kinds of VOCs were analysed using a gas chromatograph coupled with a flame ionization detector and mass spectrometer (GC-FID/MS, TSQ9000, Thermo Fisher) with a time resolution of 1 hour. The VOCs included 29 alkanes (alka), 12 alkenes (alke), 16 aromatic hydrocarbons (arom), 35 halogenated hydrocarbons (halo), 21 oxygenated volatile organic compounds (OVOCs), 1 alkyne and 1 carbon disulfide. Concentrations of NO, $NO_2$ and $O_3$ were measured employing a NOx analyser (42i, Thermo Fisher) and a $O_3$ analyzer (49i, Thermo Fisher). Data of ambient temperature (T) and relative humidity (RH) concentration were obtained from an automatic monitoring station located ~ 100 m from the container. Photolysis rate constant of $J(NO_2)$ was measured with a photolysis spectrometer (PFS-100, Focus Photonics).

An overview of observed meteorological and gas concentrations is given in Fig.9. The average temperature and relative humidity during the observation period were 25.6 °C (range from 21.9 °C to 30.8°C) and 53.8% (range from 21% to 90%), respectively. The average concentrations of NO, $NO_2$ and $O_3$ were 1.5 ppbv, 10.0 ppbv and 36.7 ppbv, respectively. Alkanes, OVOCs and alkenes were the three VOCs with the highest concentrations during the field measurement period. The corresponding average concentrations were 9.7 ppbv, 7.1 ppbv and 3.8 ppbv, respectively. The measured BVOCs only include isoprene, with an average concentration of 0.2 ppbv. The minimum and maximum values of $k_{OH}'$ were 10.6 s⁻¹ and 30.0 s⁻¹, respectively.

The reactive species and $k_{OH}'$ exhibited distinct daily variations on May 1st (the first day of the International Labour Day holiday). The peak of VOC concentrations (60 ppbv) appeared at 3:00. The concentrations of NO began increasing significantly from 3:00 onwards, reaching a peak of 23.7 ppbv around 5:00. Concurrently, the abundant NO reacted with $O_3$, resulting in the lowest observed $O_3$ concentration (~ 5 ppbv) and an increase in $NO_2$ level. $k_{OH}'$ and $NO_2$ reached their peak values of 30.0 s⁻¹ and 29.3 ppbv at about 8:00 in the morning, declining rapidly as sunlight intensified, with the lowest values observed between 12:00 and 13:00. The highest $O_3$ concentration (59.3 ppbv) occurred around 17:00 in the afternoon. $k_{OH}'$ and species exhibited relatively low values in the following days, and no significant peak in NO concentration was observed in the afternoon. These changes could be attributed to "holiday effect" (Fatahi et al., 2021), which reflect the extensive vehicle travel on the eve and the first day of the holiday and decreased human activities near observation site during the holiday (Song et al., 2022).



**Figure 9:** Time series of observed meteorological and chemical parameters, including ambient temperature, relative humidity, total OH reactivity, photolysis frequencies (J(NO₂)), and concentrations of O₃, NO, NO₂ and VOCs. The time period is from May 1st to May 4th, 2023.

A zero-dimensional box model (Framework for 0-Dimensional Atmospheric Modelling, F0AM) based on the Master Chemical Mechanism (MCM3.3.1) was applied for comparison with the observed total OH reactivity (Wolfe et al., 2016; Wei et al., 2023). The model was constrained by the measured species and parameters of meteorology and photolysis. Fig.10(a) shows the diurnal profiles of observed and simulated $k_{OH}'$. The observed $k_{OH}'$ ranged from 14.6 s$^{-1}$ (at 14:00) to 21.9 s$^{-1}$ (at 8:00) with an average value of 17.5 s$^{-1}$. According to pie chart analysis in Fig.10(b), inorganic species, alkenes, OVOCs and alkanes contributed 26.4%, 10.8%, 10.5% and 5.0% to the $k_{OH}'$, respectively. Contributions from aromatic and halogenated hydrocarbons were relatively low, accounting for 3.0% and 0.6%. The contribution of photochemical secondary





products (Others) was as high as 20.9%, while the missing reactivity (i. e., the difference between observed and simulated $k_{OH}'$) averaged ~ 20.5%, highlighting the significant role of photochemical and unidentified components in local atmospheric chemistry.

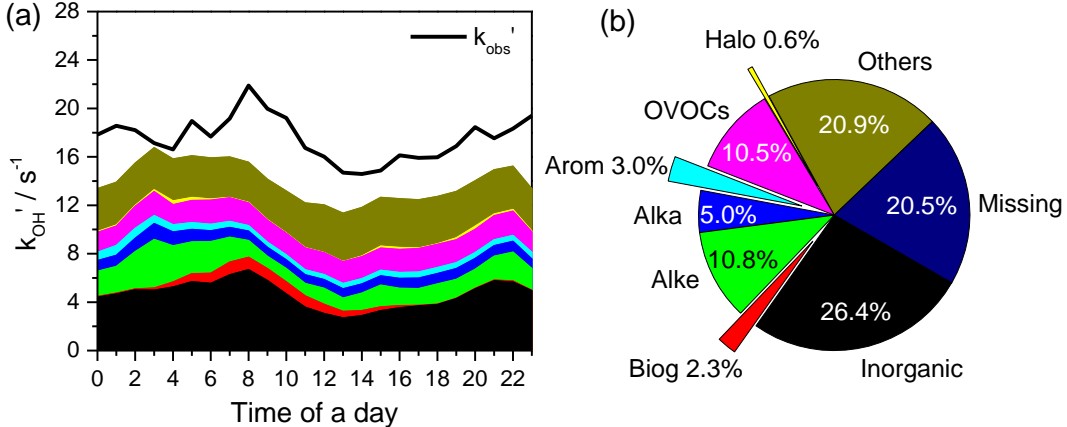

**Figure 10:** (a) Stack of diurnal profiles of observed $k_{OH}'$ compared with calculated OH reactivity from individual groups of measured atmospheric species; (b) pie chart of contributions from each groups to total reactivity.

**5 Conclusions**

A portable LP-FRS instrument was developed. A specific Herriott-type pump-probe MPC with a small multi-pass beam distribution circle radius of 3.7 mm at the centre of the cell was designed to offer an effective overlapping path length of ~ 28.4 m between the mid-infrared probe light and the photolysis light with a high overlapping factor of 75.4%. Such an overlapping factor benefit in reducing instrument size, resulting in dimensions of just 130 cm × 40 cm × 35 cm. The

375 precision and uncertainty of the LP-FRS instrument for measuring atmospheric $k_{OH}'$ were 1.0 s⁻¹ (1σ, 300 s) and within 2 s⁻¹, respectively. A Field test was performed at a suburban site, where the averaged measured $k_{OH}'$ was 17.5 s⁻¹ with a missing of 20.9% compared to the model simulated result based on measured species. The developed portable LP-FRS instrument expands the measurement capabilities for atmospheric total OH reactivity and will be employed in more field observations.

*Author contributions.* BF and WZ designed the research. BF, WZ and NY built the instrument. BF, JL and HZ conducted the instrument test. BF and HZ analysed the data. NW performed the simulation. BF and WZ wrote the paper. YL, ZZ and YL helped the field test. All authors discussed the results and commented on the paper.

*Competing interests.* The authors declare that they have no conflict of interest.





*Financial support.* This research has been supported by the National Key Research and Development Program of China (grant no. 2022YFC370030401, 2023YFC3705502), the National Natural Science Foundation of China (grant no. U21A2028, 42105099, 91544228), the 2024 Industrialization Fund of Wanjiang Emerging Industry Technology Development Centrer (grant no. WJ24CYHXM07) and the HFIPS Director's Fund (grant no. BJPY2023A02, YZJJ202101).

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
