# Peer review of "Development of a portable laser-flash photolysis Faraday rotation spectrometer for measuring atmospheric total OH reactivity"

_Atmospheric Measurement Techniques, 2024_

## Author Comment (AC1)

**Responses to the Reviewer 1**

*The authors describe the development of a novel instrument to measure total OH reactivity based on laser flash photolysis of $O_3$ in the presence of water vapour to generate OH radicals with detection of OH using Faraday rotation spectroscopy used to determine total OH loss rates in ambient air, and consequently the OH reactivity. OH reactivity is the inverse of the chemical lifetime of OH, and can be used to assess the presence and impact of unmeasured or unquantified species in ambient air, providing information regarding the production regime for secondary pollutants such as ozone. While several techniques for measuring OH reactivity have been described in the literature, measurements remain sparse and there is a need for development of alternative methods for long-term measurements.*

*The manuscript details the operating principles of the technique, and the development and characterisation of the instrument, as well as examples of initial results obtained from measurements of ambient air. The manuscript is well-written and will be of interest to the atmospheric science community. I have only minor comments, detailed below, which should be addressed prior to final publication.*

We thank the reviewer for the thoughtful and thorough reviews. Point-by-point responses to the comments are attached below. We have made corresponding modifications, and these changes are marked in the revised manuscript.

*1. Line 19: The details of the 'over-lapping factor' are probably best left to the section of the manuscript where the term is defined, the effective path length is the more significant parameter to consider (and should be 'effective' rather than 'efficient' in line 20).*

We removed the "overlapping factor" from Abstract and replaced "efficient" with "effective".

> To achieve efficient overlapping between the pump and probe laser and realize a long effective absorption path length, thus enabling high sensitivity measurement, a specific Herriott-type pump-probe optical multi-pass cell was designed. The instrument's optical box dimensions were 130 cm × 40 cm × 35 cm. The obtained effective absorption path was ~ 28.5 m in a base length of 77.2 cm.

*2. Line 25: The abstract mentions 'advantages in cost, operation, and transportation' but these are not really discussed in the manuscript, and there is no real comparison to other methods available. The costs are not mentioned at all.*

We added the information in Introduction and Sec.2 of the revised manuscript.

Revision in Introduction:
> However, the high cost of development and operation (e. g. the expensive and complex dye laser system and mass spectrometer system), limited instruments, complex operation and calibration procedures, and relatively large size of these instruments hinder the widespread application of measuring OH reactivity.

> The time-resolved LP-FRS is a novel technique that employs a mid-infrared semiconductor diode laser (with much cheaper commercial price than the dye laser system and good stability) as the probe laser for $k_{OH}$' measurement, making the technique both cost-effective and simple to operate (Wei et al., 2020).

Revision in Sec.2:
> Optical components from both systems are integrated into a single unitary box, with all communications and gas tubes connected to designated interfaces. The optical box has dimensions of 130 cm × 40 cm × 35 cm and a total weight of ~ 90 kg. The instrument's total operation power consumption is ~ 3 kW. These factors make the developed LP-FRS instrument both cost-effective and portable for field applications.

*3. Line 61: The statement 'without needing to determine the reaction time' is a little confusing, knowledge of the reaction time is essential.*

We added the knowledge of the reaction time in the manuscript to clarify.

> The LP-LIF is a pump-probe technique where OH decay can be observed with high time resolution after each flash, without needing to determine the reaction time from the point of OH production to the sampling position. In this technique, OH is produced by laser-flash photolysis of $O_3$ at 266 nm across the entire illuminated area in the presence of water vapour.

*4. Lines 64-65: It would be helpful to provide a brief explanation of the problems at high NO concentrations in instruments using photolysis of water vapour to produce OH, and how measurements of $H_2SO_4$ provide information on OH.*

We added the two contents in the revised manuscript.

In this technique, OH is produced by laser-flash photolysis of $O_3$ at 266 nm across the entire illuminated area in the presence of water vapour. This makes it less susceptible to the recycling process caused by nitric oxide (NO) compared to the above instruments using water vapour photolysis (Sadanaga et al., 2004; Lou et al., 2010). Because water vapour photolysis with 184.9 nm UV lamp not only generates OH but also produces $HO_2$ radicals. In the presence of high atmospheric NO concentrations, the reaction of $HO_2$ with NO can lead to the reformation of OH, which may affect the measurement of $k_{OH}'$.

In the semi-direct technique of FT-CIMS, sulphuric acid ($H_2SO_4$) instead of OH is measured by a CIMS instrument to record the data point of OH decay at each reaction time. The reaction time can be varied by adding 10-ppmv $SO_2$ at different fixed positions within the flow tube. Due to the titration reaction, OH is nearly completely converted to $H_2SO_4$, so the measured change in $H_2SO_4$ concentration serves as an indicator of the OH.

*5. Line 68: A brief summary of the main conclusions of the work by Fuchs et al. would be helpful.*

We added the brief conclusions in the manuscript.

The result shown that the indirect or semi-direct methods exhibited more scattered in measurements and are most likely limited by the corrections for known effects, such as high NO concentrations for CRM and high reactivity conditions for FT-CIMS. In comparison, the direct methods (LIF) that combine laser-flash photolysis offer advantages in detection precision and accuracy. Overall, the existing techniques can give reasonable measurement results for a wide range of atmospheric conditions. However, the high cost of development and operation (e. g. the expensive and complex dye laser system and mass spectrometer system), limited instruments, complex

operation and calibration procedures, and relatively large size of these instruments hinder the widespread application of measuring OH reactivity.

*6. Line 103: What does the line strength equate to in terms of a cross-section under the operating conditions of the experiment?*

The relationship between cross-section and line strength can be expressed as $\sigma(v) = S \times g(v)$ (Chen et al., Photonic sensing of reactive atmospheric species, John Wiley & Sons, 2017). Where $g(v)$ is the lineshape. At operating conditions of 200 mabr and 298 K, the value of Voigt lineshape is $g(v=3568.523 \text{ cm}^{-1})=15.32$. Thus, the line strength is 1/15.32 of the cross-section.

*7. Line 154: 'angel' to 'angle'.*

Done.

*8. Table 1 and discussion lines 168-173: The comparison of overlapping factors seems a little unnecessary, and perhaps misleading. The papers described measure in different regions of the spectrum, where absorption cross-sections are likely higher, and so have less need for the longer effective path lengths developed for the measurements described in the manuscript. It would be more beneficial to provide a comparison of limits of detection.*

The limit of detection is indeed crucial, and a key factor influencing it is the effective absorption path length of the optical system used. In this regard, Table 1 illustrates our efforts to extend the optical path length to improve the detection limit, while also reducing the size of the LP-FRS instrument. The overlapping factor reflects both the efficiency of optical path length utilization and the compactness of the pump-probe Herriott cell. Therefore, the comparison presented in Table 1 is essential. To further clarify the point, we revised the description accordingly.

**Table 1.** Overlapping factor comparison of MPCs used for laser photolysis in pump-probe techniques.

In fact, from an optical structure perspective, the overlap factor characterizes both the utilization efficiency of the optical path length and the compactness of the pump-probe MPC, making it an ideal parameter for performance characterization. To demonstrate the efforts in reducing instrument size, a

comparison of effective overlapping path lengths and overlapping factors with literature reported pump-probe MPCs is shown in Table 1.

*9. Line 190 (and elsewhere): It would be better to be consistent throughout with use of µV and nV.*

The unit of noise in the manuscript were modified to nV Hz$^{-1/2}$.

*10. Line 246: It would be better to give the equation in terms of the concentration (or signal) of OH.*

We revised the terms of Eq.6 and the description.

$$S_t = S_{backgroud} + S_0 \exp(-k_{decay}t) \tag{6}$$

where $S_0$ and $S_t$ are the FRS signal intensities proportional to OH concentration at the time when the fitting started and at the time $t$, respectively. $S_{backgroud}$ is the background signal intensity.

*11. Line 249: It would be better to give the time since photolysis in place of 'the 180$^{th}$ data point'.*

We revised "the 180$^{th}$ data point" to "the time of 36 ms".

the fit is started at the time of 36 ms rather than the peak to avoid any fluctuations affecting the fitting result

*12. Figure 5: Time zero is more commonly described as the point at which photolysis occurs.*

Yes, time zero is commonly described as the point at which photolysis occurs, such as the time sequence in LP-LIF instrument. But in our system, as shown in the following figure, timing is started from the data acquisition to record first 30-ms data points to evaluate noise level and perform rapid background subtraction to obtain clear OH spectral signal for determining the operating current of the laser.

[Figure]

Fig.1: Time sequence of the developed portable LP-FRS instrument

*13. Line 262: Please reformat the equations to express the uncertainties more clearly.*

Done.

The measured values are agreement with the IUPAC (International Union of Pure and Applied Chemistry) recommend values of $6.4_{+1.3}^{-1.1} \times 10^{-15}$ cm³ molecule⁻¹ s⁻¹, $1.6_{+0.2}^{-0.2} \times 10^{-13}$ cm³·molecule⁻¹ s⁻¹, and $3.0_{+0.8}^{-0.6} \times 10^{-12}$ cm³·molecule⁻¹ s⁻¹, respectively (Atkinson et al., 2004; Atkinson et al., 2006).

*14. Figure 6: Please give the equation for the line in terms of physical parameters (x is also missing in panel a), and add lines representing the literature values for the rate coefficients to the plots for comparison.*

We gave the fitting equation in terms of physical parameters in the main text and Fig.6. And after consideration, we believe that providing the reference values directly in the caption of Fig.6 can better illustrate the experimental results compared to adding reference lines for the rate coefficients.

The OH decay rates in the reactions with three different species were measured and can be expressed as $k_{\text{decay}} = k_{\text{OH+X}}[\text{X}] + k_0$. Where $k_{\text{OH+X}}$ is

the measured rate constant for the reaction of OH with X, [X] is concentration of reactant X, $k_0$ is a background value.

[Figure]

**Figure 6:** Plots of the measured pseudo-first-order rate coefficients vs (a) $CH_4$ concentrations, (b) CO concentrations and (c) NO at 298 K. The measured reaction rate constants which obtained from the slopes are $k_{OH+CH_4} = 6.4^{-0.1}_{+0.1} \times 10^{-15}$ cm³ molecule⁻¹ s⁻¹, $k_{OH+CO} = 1.6^{-0.1}_{+0.1} \times 10^{-13}$ cm³ molecule⁻¹ s⁻¹, and $k_{OH+NO} = 3.0^{-0.1}_{+0.1} \times 10^{-12}$ cm³ molecule⁻¹ s⁻¹, which respectively agree with the recommend values of $6.4^{-1.1}_{+1.3} \times 10^{-15}$ cm³ molecule⁻¹ s⁻¹, $1.6^{-0.2}_{+0.2} \times 10^{-13}$ cm³·molecule⁻¹ s⁻¹, and $3.0^{-0.6}_{+0.8} \times 10^{-12}$ cm³·molecule⁻¹ s⁻¹, respectively.

*15. Line 279: How does the correction factor impact the uncertainties of the reactivity measurements?*

The reactivity measurement uncertainty introduced by the correction factor is about 2%, due to the high accuracy of the MFCs and pressure controller.

*16. Line 262 (and elsewhere): Please provide the uncertainties for measured rate coefficients.*

Done.

As shown in Fig.6, the obtained reaction rate constants for OH + $CH_4$, OH + CO, and OH + NO at 298 K were found to be $k_{OH+CH_4} = 6.4^{-0.1}_{+0.1} \times 10^{-15}$ cm³ molecule⁻

$^{1}$ s$^{-1}$,  $k_{\mathrm{OH+CO}} = 1.6^{-0.1}_{+0.1} \times 10^{-13}$ cm$^3$ molecule$^{-1}$ s$^{-1}$, and  $k_{\mathrm{OH+NO}} = 3.0^{-0.1}_{+0.1} \times 10^{-12}$ cm$^3$

molecule$^{-1}$ s$^{-1}$, respectively.

*17. Line 326: A table summarising the species measured and mean/median concentrations would be helpful.*

The mean and median concentrations of measured reactants during the observation period was given in Table 2 in the form of classification. The description in the main text was also modified accordingly.

Table 2 summarizes the mean and median concentrations of the measured species during the observation period. The mean concentrations of NO, NO$_2$ and O$_3$ were 1.5 ppbv, 10.0 ppbv and 36.7 ppbv, respectively. Alkanes, OVOCs and hydrocarbons were the three VOCs with the highest concentrations during the period.

**Table 2.** The mean and median concentrations of measured species during the observation period in the form of classification.

| Measured species | Average concentrations (ppbv) | Median concentrations (ppbv) |
|---|---|---|
| NO | 1.5 | 0.3 |
| NO$_2$ | 10.0 | 8.9 |
| O$_3$ | 36.7 | 36.9 |
| alka | 9.1 | 9.0 |
| alke | 3.8 | 3.1 |
| arom | 2.3 | 2.2 |
| halo | 4.9 | 4.6 |
| BVOCs (only include isoprene) | 0.2 | 0.1 |
| OVOCs | 7.1 | 7.1 |

*18. Figure 9: There are large changes in J(NO$_2$) throughout the measurement period, including one day when it is near-zero at midday. Is there an explanation for this variation?*

The weather was the cause of the large changes in J(NO$_2$). We added the explanation in the revised manuscript.

An overview of observed meteorological and gas concentrations is given in

Fig.9. The average temperature and relative humidity during the observation period were 25.6 ℃ (range from 21.9 ℃ to 30.8 ℃) and 53.8% (range from 21% to 90%), respectively. The large changes in $J(NO_2)$ were due to the raining weather on May 3rd and cloudy conditions on May 4th.

---

## Author Comment (AC2)

**Responses to the Reviewer 2**

*The present manuscript describes in detail a new experimental technique developed for measuring total OH reactivity. The instrument is compact and can easily be deployed in field campaigns. The compact size has benefits compared to currently used instruments and will possibly become a standard instrument, also for long-term measurements.*

*The paper is very well written, with many details on the instrumental design and the validation. I have no major remarks, only a few minor comments that might improve the manuscript:*

We thank the reviewer for the thoughtful and thorough reviews. Point-by-point responses to the comments are attached below. We have made corresponding modifications, and these changes are marked in the revised manuscript.

*1. Line 67: There has been another intercomparison campaign between CRM and LP-LIF technique, which you might want to cite for completeness: Hansen et al., Intercomparison of the Comparative Reactivity Method (CRM) and Pump-Probe technique for measuring total OH reactivity in an urban environment, AMT 8, 4243 (2015)*

We added the work and revised the description in the manuscript.

> Several intercomparisons of the techniques and instruments mentioned above have been conducted. Zannoni et al. (2015) reported a field intercomparison of two CRM instruments in the Mediterranan basin. Hansen et al. (2015) carried out an intercomparison between the CRM and LP-LIF techniques in an urban environment. The series of intercomparison experiments conducted in the SAPHIR simulation chamber at Forschungszentrum Jülich involved all types of existing methods and nine instruments from around the world (Fuchs et al., 2017).

*2. Line 113: demodulated*
*3. Line 123: spots ARE arrangeD*
*4. Line 135: consistsING*
*5. Line 145: undergoesING*
*6. Line 156: below than 79.2*
*7. Line 169: several*

8. Line 199: affecting
9. Line 205: correspondingS
10. Line 222: producingED
11. Line 241: were was
12. Line 245: flowing following

We made the modifications based on Suggestions 2 to 12.

13. *Figure 5: It would be more convenient to define as time 0 the time when the photolysis laser is triggered.*

Yes, time zero is commonly defined as the point at which laser photolysis is triggered, such as the time sequence in LP-LIF instrument. But in our system, as shown in the following figure, timing is started from the data acquisition to record first 30-ms data points to evaluate noise level and perform rapid background subtraction to obtain clear OH spectral signal for determining the operating current of the laser.

[Figure]

Fig.1: Time sequence of the developed portable LP-FRS instrument

14. *Line 290: You consider that the remaining around 80% of the $k_{zero}$ is due to reaction with impurities?*

Yes, the impurities in additional zero flow and water are considered as the major contribution to the instrument zero.

*15. Figure 7: How k_OH can be negative? I guess you removed the overall average value (5.2s⁻¹?) from each individual value? Please specify.*

Yes, the measurement is the correction value after being processed with instrument zero and dilution, as described in Eq.7 of $k'_{OH} = f \times (k_{decay} - k_{zero})$. We added the description in the manuscript.

> To assess the measurement precision of the LP-FRS instrument, an Allan deviation analysis was conducted on time series of zero air measurements. The measured values were processed by subtracting the instrument zero.

*16. Line 299: with A total a*

Done.

*17. Line 300: Can you confirm that the measurement precision is already converted to atmospheric conditions, ie. that the precision of the measurement itself is 5 times better?*

Yes, we confirm the precision is converted to atmospheric conditions. The measured values in Fig.7 were processed with the instrument zero and the dilution factor.

*18. Line 302: improved*

Done.

*19. Figure 8: same question: how can k_OH be negative? And what is the mean value of 0.2 s⁻¹?*

The measured values have been processed by subtracting the instrument zero. The value of 0.2 s⁻¹ reflects the relatively stability of multiple long interval measurements, without any significant measurement drift.

*20. Line 329: (RH) concentration*
*21. Line 376: missing reactivity of*

Done. We made the modifications based on Suggestions 20 to 21.

*22. Finally, some more information on energy consumption, total weight, and also an indication on the cost of the instrument could be interesting to the reader.*

We added the information at appropriate locations in the revised manuscript.

Revision in Introduction:

The time-resolved LP-FRS is a novel technique that employs a mid-infrared semiconductor diode laser (with much cheaper commercial price than the typical dye laser system and good stability) as the probe laser for $k_{OH}'$ measurement, making the technique both cost-effective and simple to operate (Wei et al., 2020).

Revision in Sec.2:

Optical components from both systems are integrated into a single unitary box, with all communications and gas tubes connected to designated interfaces. The optical box has dimensions of 130 cm × 40 cm × 35 cm and a total weight of ~ 90 kg. The instrument's total operation power consumption is ~ 3 kW. These factors make the developed LP-FRS instrument both cost-effective and portable for field applications.

---

## Author Comment (AC3)

**Responses to the Reviewer 3**

*Quantitative measurements of total OH reactivity ($k_{OH}$') provide important insights into atmospheric photochemistry. This paper reports the development of a portable LP-FRS instrument for real-time and in-situ measurement of $k_{OH}$'. The size of the instrument is reduced to 130×40×35 cm. Its advantages in cost, operation, and transportation make it of great value in field observation and laboratory research. The optical and mechanical structure, the key MPC subsystem, parameter optimization, laboratory performance analysis, and field operation demonstration of the instrument are introduced in detail. I recommend acceptance after considering the following minor comments:*

We thank the reviewer for the thoughtful and thorough reviews. Point-by-point responses to the comments are attached below. We have made corresponding modifications, and these changes are marked in the revised manuscript.

*1. Attention should be paid to textual details, such as the missing "x" in Fig.6(a), line 159 (missing $r_c$ unit), and the inconsistencies of time resolution value between line 299 and caption of Fig.7(a).*

Done. We checked the manuscript and revised textual errors.

*2. Table 1 does not seem to show the performance, but rather the compactness of pump-probe multi pass cells. It would be better to make a modification.*

For a pump-probe optical system, the effective absorption path length and overlapping factor are the best indicators of the system performance. Meanwhile, the overlapping factor reflects the system compactness which benefit in reducing instrument size. Therefore, we made the comparison presented in Table 1. To further clarify the point, we made modifications in the manuscript.

> **Table 1.** Overlapping factor comparison of MPCs used for laser photolysis in pump-probe techniques.
>
> In fact, from an optical structure perspective, the overlap factor characterizes both the utilization efficiency of the optical path length and the compactness of the pump-probe MPC, making it an ideal parameter for performance characterization. To demonstrate the efforts in reducing instrument size, a

comparison of effective overlapping path lengths and overlapping factors with literature reported pump-probe MPCs is shown in Table 1.

*3. What is model of the UV lamp used to generate O$_3$?*

We added the model to the manuscript.

> To produce sufficient OH in the MPC, a small flow rate (~ 0.25 L/min) of zero air which passed through a UV lamp (UVP PenRay, Analytikjena) and a bubbling bottle to generate O$_3$ and water vapour, is added to the main sampling flow (~ 6.0 L/min), resulting in a total flow rate of 6.25 L/min.

*4. Adding some explanations to the beginning of line 67 of the Introduction section will be more helpful in explaining the problems of the current methods and instruments.*

Done. At the beginning of this paragraph, we introduced several intercomparison experiments and then made a brief summary to describe the common limitations and advantages of current techniques and instruments.

> Several intercomparisons of the techniques and instruments mentioned above have been conducted. Zannoni et al. (2015) reported a field intercomparison of two CRM instruments in the Mediterranan basin. Hansen et al. (2015) carried out an intercomparison between the CRM and LP-LIF techniques in an urban environment. The series of intercomparison experiments conducted in the SAPHIR simulation chamber at Forschungszentrum Jülich involved all types of existing methods and nine instruments from around the world (Fuchs et al., 2017). The result shown that the indirect or semi-direct methods exhibited more scattered in measurements and are most likely limited by the corrections for known effects, such as high NO concentrations for CRM and high reactivity conditions for FT-CIMS. In comparison, the direct methods (LIF) that combine laser-flash photolysis offer advantages in detection precision and accuracy.

*5. The role of r$_c$ is clear because it directly affects the overlap. However, the effect of g value is not obvious and needs to be briefly explained.*

We added the description of *g* value.

where $g$ ( $g = g_1 = g_2 = \cos\theta = 1 - d / R$ ) is the parameter that describes the optical resonance stability of optical cavity or MPC. When the $g$ value is less than zero, the base length of the multi-pass cell exceeds the curvature radius of the mirror, causing the resonance to become unstable. An incident parallel beam or a beam whose waist is not properly aligned with the cell's centre will quickly diverge and cannot be collected. However, a beam with its waist well matched to the centre can effectively prevent divergence. $\theta$ is half of the angle between two adjacent reflection light points on mirror surface. $d$ is the base length, $R_1 = R_2 = R$ is the curvature radii of the mirrors. $r$ is the radius of the spot distribution circle.

*6. Why is the error bar of OH+NO in Fig.6 larger than the other two experiments?*

This is due to two reasons. One is that the measurement averaging time is different from the other two experiments. The other may be that the reaction rate of NO+OH is very fast. The flow fluctuations can cause larger measurement fluctuations.

*7. Line 261. The expression of "the slope errors of the fittings were less than 0.1" is not clear.*

We directly put the slope errors into the reaction rate constant values in the revised manuscript to clarify the expression.

As shown in Fig.6, the obtained reaction rate constants for OH + CH₄, OH + CO, and OH + NO at 298 K were found to be $k_{OH+CH_4} = 6.4^{-0.1}_{+0.1} \times 10^{-15}$ cm³ molecule⁻¹ s⁻¹, $k_{OH+CO} = 1.6^{-0.1}_{+0.1} \times 10^{-13}$ cm³ molecule⁻¹ s⁻¹, and $k_{OH+NO} = 3.0^{-0.1}_{+0.1} \times 10^{-12}$ cm³ molecule⁻¹ s⁻¹, respectively.

*8. Line 312. The residual reactants in water are also one of the sources of uncertainty affecting the zero determination of the instrument.*

We added the uncertainty to the manuscript.

The uncertainty of the instrument zero primarily originates from various influencing factors, such as changes in O₃ concentrations and residual reactants in zero air and water.